# In Vitro Activities of MMV Malaria Box Compounds against the Apicomplexan Parasite *Neospora caninum*, the Causative Agent of Neosporosis in Animals

**DOI:** 10.3390/molecules25061460

**Published:** 2020-03-24

**Authors:** Joachim Müller, Pablo A. Winzer, Kirandeep Samby, Andrew Hemphill

**Affiliations:** 1Department of Infectious Diseases and Pathobiology, Institute of Parasitology, Vetsuisse Faculty, University of Bern, Länggass-Strasse 122, 3012 Bern, Switzerland; 2Graduate School for Cellular and Biomedical Sciences, University of Bern, Mittelstrasse 43, 3012 Bern, Switzerland; pablo.winzer@vetsuisse.unibe.ch; 3Medicines for Malaria Venture (MMV), 20, Route de Pré-Bois, 1215 Geneva 15, Switzerland; sambyk-consultants@mmv.org

**Keywords:** antiparasitic chemotherapy, mitochondrion, mode of action, screening, target

## Abstract

(1) Background: *Neospora caninum* is a major cause of abortion in cattle and represents a veterinary health problem of great economic significance. In order to identify novel chemotherapeutic agents for the treatment of neosporosis, the Medicines for Malaria Venture (MMV) Malaria Box, a unique collection of anti-malarial compounds, were screened against *N. caninum* tachyzoites, and the most efficient compounds were characterized in more detail. (2) Methods: A *N. caninum* beta-galactosidase reporter strain grown in human foreskin fibroblasts was treated with 390 compounds from the MMV Malaria Box. The IC_50_s of nine compounds were determined, all of which had been previously been shown to be active against another apicomplexan parasite, *Theileria annulata*. The effects of three of these compounds on the ultrastructure of *N. caninum* tachyzoites were further investigated by transmission electron microscopy at different timepoints after initiation of drug treatment. (3) Results: Five MMV Malaria Box compounds exhibited promising IC_50_s below 0.2 µM. The compound with the lowest IC_50_, namely 25 nM, was MMV665941. This compound and two others, MMV665807 and MMV009085, specifically induced distinct alterations in the tachyzoites. More specifically, aberrant structural changes were first observed in the parasite mitochondrion, and subsequently progressed to other cytoplasmic compartments of the tachyzoites. The pharmacokinetic (PK) data obtained in mice suggest that treatment with MMV665941 could be potentially useful for further in vivo studies. (4) Conclusions: We have identified five novel compounds with promising activities against *N. caninum*, the effects of three of these compounds were studies by transmission electron microscopy (TEM). Their modes of action are unknown and require further investigation.

## 1. Introduction

Apicomplexan parasites, in particular *Neospora*, *Toxoplasma*, *Cryptosporidium*, *Eimeria*, *Plasmodium*, and *Theileria*, are important pathogens that affect humans and/or animals. *Neospora caninum*, which is closely related to *Toxoplasma gondii*, is a major causative agent of abortion in cattle and causes neuromuscular disease in dogs and other domestic animals [1].

The definitive hosts of *N. caninum* are canids, in particular dogs with a worldwide prevalence of 17%, as estimated in a recent meta-review [2]. Besides cattle, sheep, water buffaloes, and many wildlife species can act as intermediate hosts [3]. Since the economic losses due to abortion in cattle are high, amounting up to 1.3 billion US dollars per year according to data obtained from 10 countries [4], several strategies are considered for the prevention and treatment of bovine neosporosis, namely (i) the testing and culling of seropositive animals, (ii) discontinued breeding with offspring from seropositive cows, (iii) the vaccination of susceptible and infected animals, and (iv) the chemotherapeutical treatment of calves from seropositive cows [5]. Strategies (iii) and (iv) are, however, economically viable only if suitable targets and effective formulations for vaccination and/or drug treatments are identified [6].

Ab initio drug development against neosporosis is certainly not possible due to the high costs and low market return compared to other diseases. Consequently, the repurposing of compounds effective against other pathogens may constitute a suitable approach. Despite the fact that a wide range of compound classes exhibit interesting effects against *N. caninum* tachyzoites in vitro [6], only a few have been demonstrated to be effective in suitable animal models. One example is buparvaquone (BPQ), a drug effective against bovine theileriosis [7], currently marketed in Africa. BPQ is effective against *N. caninum* in vitro and in vivo [8], and inhibits the vertical transmission of *N. caninum* [9] and of *T. gondii* [10] in pregnant mouse models.

In 2011, to stimulate drug discovery beyond malaria, Medicines for Malaria Venture (MMV) created the open-source MMV Malaria Box issuing from three previous screenings of libraries from the St. Jude hospital [11,12], from GlaxoSmithKline [13], and from Novartis [14,15]. Further selections resulted in 200 drug-like and 200 probe-like compounds with IC_50_s against *Plasmodium falciparum* blood stages below 4 µM and a more than 10 times lower cytotoxicity against HEK-293 cells [16].

The Malaria box was freely accessible until 2016. Data on activities, structures, chemical properties, cytotoxicity are, however, still openly available (www.mmv.org/mmv-open) and in vitro absorption distribution-metabolism-excretion (ADME) and in vivo pharmacokinetic data is available on request. Consequently, preclinical and clinical candidates for a wide range of diseases have been identified. The box has been tested against a wide range of pathogens, including apicomplexan parasites of medical and economic importance, and the outcome of these studies has been published by Van Voorhis et al. [17]. Since that particular study has focused mainly on human diseases, the effects on parasites with a merely veterinarian importance, such as *N. caninum* or *Theileria* sp. have not been extensively covered. Thus, we have screened the Malaria box against *N. caninum* and *T. annulata* in separate studies, and the results of the screening against *T. annulata* were published elsewhere [18]. Here, we present the results of a differential screening of the Malaria Box against *N. caninum* and their host cells used for in vitro maintenance of these parasites. Nine compounds effective against *N. caninum* and *T. annulata*, but not against host cells were characterized in more detail.

## 2. Results

### 2.1. Differential Screening

In a first differential screening, the 390 compounds contained in the MMV Malaria Box were screened against human foreskin fibroblasts (HFFs) infected with a *N. caninum* strain expressing *Escherichia coli* β-galactosidase to determine the effectiveness against *N. caninum* tachyzoites and against uninfected HFFs to determine cytotoxicity. Thirty-nine compounds were effective against *N. caninum*. eleven of these compounds were, however, cytotoxic against HFFs. Of the resulting twenty-eight compounds (19 were probe-like, nine were drug-like) with specific antiparasitic activity, nine were also effective against *Theileria annulata* (Figure 1). The MMV numbers of the 28 “Hits” are given in Table A1 in Appendix A.

### 2.2. Inhibition Curves

In a next step, the IC_50_ values of the nine compounds effective against both, *N. caninum* and *T. annulata*, were determined using a concentration range of 0–250 nM. Four compounds (1–4) had IC_50_ values below 0.1 µM, and one compound (5) had IC_50_ values below 0.2 µM (Figure 2A). The IC_50_ values of the four compounds 6–9 were above the concentration range used (Figure 2B) and had to be extrapolated.

Compound (1), MMV665941, had the lowest IC_50_ with 25 nM. Effectiveness and LogP were not correlated since lipophilic compounds, e.g., compounds (1–4), and a hydrophilic compound (5) had IC_50_ values below 0.2 µM (Table 1).

The structures of the most effective compounds (1–5) are depicted in Figure 3.

Interestingly, the structure of compound (1) is very similar to crystal violet (which lacks the OH group). However, structurally and in terms of its properties, this compound is clearly different from crystal violet.

### 2.3. Effects on the Ultrastructure of *N. caninum* Tachyzoites

In a next step, we investigated the effects of three of these five compounds, namely compound (1), compound (2), compound (5), and dimethyl-sulfoxide (DMSO) as a control, on the parasite ultrastructure. The selection of test compounds included the two compounds with the lowest IC_50_ (1 and 2), and compound (5), which had an unusually low logP (Table 1).

In control cultures, *N. caninum* tachyzoites were located intracellularly, localized within a parasitophorous vacuole (PV) that is delineated by a parasitophorous vacuole membrane (PVM), which is essentially host cell surface membrane-derived and then modified by the parasites. Tachyzoites proliferate within the PV, and thus the number of parasites within the vacuoles increased rapidly with time (see Figure 4). Within the PV, tachyzoites were embedded in a matrix, composed of the parasitophorous vacuole tubular network (PVN). Structural features of tachyzoites were easily identified, including the apical conoid, and secretory organelles including dense granules, micronemes, and the rhoptries. Also discernible were the nucleus, the Golgi apparatus, and the mitochondrion with an electron-dense matrix (Figure 4).

In cultures treated with compound (1), the number of parasites per vacuole did not notably increase compared to the controls no matter what time point was studied, reflecting the significant inhibition of proliferation (Figure 5). In addition to a profound inhibition of proliferation, compound (1) also induced structural alterations within the tachyzoites, such as the cytoplasmic vacuoles that were already detectable after 6 h of treatment. These vacuoles appeared to originate from the mitochondrion, which has lost its electron dense matrix and was not visible anymore (Figure 5A,B). At this timepoint, the PVM was not always clearly detectable (Figure 5B). The overall degeneration of the tachyzoites progressed with time, vacuolization in the parasite cytoplasm became more profound (Figure 5C), and after 48 h of drug treatment, tachyzoites were either completely degenerated (Figure 5E) or the cytoplasm was filled with electron-dense material and translucent inclusions, which are most likely amylopectin granules (Figure 5D,F). The host cell cytoplasm, the nuclei and also the mitochondria appeared to remain unaffected by the treatment with compound (1).

In cultures treated with compound (2), only a few alterations, such as vacuolization, lamellated membrane, and stacks in some instances could be observed after 6 h of treatment, but the mitochondrial matrix remained intact, indicating a slightly delayed effect compared to compound (1) (Figure 6A,B). However, after a treatment duration of 12 h (Figure 6C), the loss of the tachyzoite mitochondrial matrix was evident, and residues of cristae were still visible on some vacuolar membranes. At 24 h of treatment, a subset of parasites started to exhibit a electron-dense cytoplasm, while others still displayed intact structures despite vacuolization (Figure 6D). After 48 h, parasites were often seen residing within an electron dense matrix, and containing large numbers of amylopectin granules (Figure 6E). The host cells remained largely unaffected.

In cultures treated with compound (5), tachyzoites forming daughter zoites emerging upon endodyogeny were visible throughout the whole experiment. After 6 h and even after 12 h, we detected vacuoles with numerous zoites that were either vacuolized or exhibited a fragmented cytoplasm, all of which indicated ongoing proliferation despite clear ultrastructural alterations. At 48 h of treatment, large vacuoles were observed, exhibiting an electron-dense matrix containing numerous parasites with altered ultrastructures. Frequently, completely destroyed tachyzoites, still enclosed by a vacuole, were observed at this time point (Figure 7).

## 3. Discussion

An in vitro screening of 390 MMV Malariabox compounds against *Neospora caninum* yielded in the identification of 28 compounds with specific activity against *N. caninum*. Nine of these compounds were also effective against *Theileria annulata* [18], another apicomplexan parasite affecting cattle, transmitted by ticks. Of these nine compounds, the most effective is compound (1) or MMV665941, a triphenyl-methanol closely related to crystal violet. MMV665941 was previously also shown to be most effective against *T. annulata* but also clearly toxic for macrophages with a therapeutic index of only 2.5 [18]. The fact that, in the present screening, MMV665941 did not affect confluent HFFs could indicate that this compound preferentially affects proliferating cells. Moreover, MMV665941 was also shown to be moderately effective against *Cryptosporidium parvum* with an IC_50_ of around 0.8 µM [19]. In a recently published study comparing the rapidity of action of MMV Malaria Box compounds against *Plasmodium falciparum* blood forms with the (potential) mode of antimalarial action [20], MMV665941 was designated to act more slowly than artesunate, but faster than atovaquone. The mechanism of action of this compound is unknown. Studies on MMV665941-susceptible yeast strains suggested an interference with sterol-dependent processes since two of seven susceptible mutants were defective in sterol biosysnthesis [17].

The closely related crystal violet is a broad-spectrum antiseptic with multiple cellular targets. It binds to bacterial cell wall components, to DNA [21], most likely also to other acidic macromolecules [22], and uncouples *Trypanosoma* mitochondria [23]. These multiple targets may explain why this compound has not only the lowest IC_50_, but also the strongest impact on parasite ultrastructure, as shown by transmission electron microscopy (TEM). Since the vacuolization of tachyzoites observed already after 6 h could correspond to the degradation of mitochondria, the selectivity of MMV665941 could be due to a preferential action on parasite vs. host mitochondria. In terms of the pharmacokinetic properties in mice, MMV665941 appears as the most promising compound of the three, due to a maximal plasma concentration (C_max_) of 1.09 µM and a high value for the area under the curve from time point 0 to the time point of the last measurable concentration (AUC_0-last_) (see Table A2).

Interestingly, also in cultures treated with compounds (2) and (5), intact parasite mitochondria are seen only at 6 h after treatment, not later. This may indicate that compounds with different structures and chemical properties (note the different logP values!) may affect the integrity of apicomplexan mitochondria. For instance, endochin-like quinolones affect the ultrastructure of mitochondria—as evidenced by digestive vacuoles containing mitochondrial fragments—but not of other organelles in *N. caninum* tachyzoites [24]. In addition, organometallic ruthenium complexes were shown to primarily act on mitochondria in *Toxoplasma gondii* [25], *N. caninum* [26], and in the extracellular protozoan parasite *Trypanosoma brucei* [27].

Compound (2) or MMV665807 is a trifluorinated salicylanilide that acts against *P. falciparum* blood stages very rapidly [20]. For similar compounds, the predicted target is PfATP4, an Na+-ATPase of the plasma membrane [28,29]. The target in *N. caninum* is unknown. According to our TEM observations at 6 h after treatment, the action of MMV665807 against the parasite seems, however, slower than the action of MMV665941, indicating that extrapolations of the action of antimalarials against blood forms of *P. falciparum* to intracellular stages of other apicomplexan parasites should be made with caution.

Compounds (3) and (4) are both 8-hydroxyquinolines. Compound (3), MMV000787, acts very slowly against *P. falciparum* blood forms [20] and inhibits cysteine proteinases in functional assays [17]. Compound (4), MMV666054, is also most effective against *T. annulata*, but with a relatively low cytotoxicity against bovine macrophages yielding a therapeutic index of around 18 [18]. Moreover, MMV666054 is effective against *C. parvum* [19] and against the helminth *Brugia malayi* [17]. MMV666054 acts relatively slowly against *Plasmodium falciparum* blood forms, most likely by interfering with heme catabolism [20]. Another potential target is MAP kinase 2, as revealed by functional assays with *Leishmania donovani* MAPK2 [17].

Compound (5), the naphthalenediimide MMV009085, is the only hydrophilic compound with a submicromolar activity (IC_50_ = 150 nM) against *N. caninum*. It is also effective against *C. parvum* with an IC_50_ in same range [19]. This compound potentially inhibits hexose uptake, as evidenced by selective uptake inhibition studies involving *P. falciparum* hexose transporter 1 [30]. MMV009085 is hydrophilic and may directly interact with the transporter as a hexose analogue. While there are no studies on hexose transporters in *N. caninum*, it has been shown that the closely related *T. gondii* expresses two plasma membrane-associated hexose transporters (TgGT1 and TgST2), and two intracellularly localized transporters (TgST1 and TgST3). Surprisingly, TgGT1 and TgST2 are nonessential to the parasite, but deletion of TgGT1 and TgGT1/TgST2 was shown to cause a delay in intracellular replication, the effect of which could be abolished by adding glutamine as a complement substrate [31]. Thus, hexose transporters are nonessential for *T. gondii*, and possibly for *N. caninum*, and other mechanisms of action could be responsible for the observed effects upon treatment with compound (5) in *N. caninum* tachyzoites.

In conclusion, we have identified five MMV Malaria box compounds that inhibit the proliferation of *N. caninum* tachyzoites in vitro with IC_50_s in the range of 25-150 nM. Three of these compounds were further studied with respect to their effects on morphology and ultrastructure, confirming that they do not only inhibit proliferation but also cause extensive alterations, most notably in the parasite mitochondria but also in other cytoplasmic compartments within these parasites. How these compounds exert these effects is not known, and respective studies on modes of action in *N. caninum* will require further investigations. In any case, these compounds are interesting candidates for follow-up studies in vivo, especially in animal models that allow for the monitoring of the vertical transmission of *N. caninum*, which still remains a major problem that causes substantial economic losses.

## 4. Materials and Methods

### 4.1. Chemicals

The MMV Malaria Box contains 390 compounds (instead of 400, as initially planned) in the form of 10-mM stock solutions in DMSO and was obtained free of charge from the MMV (Geneva, Switzerland). Supporting information, including the plate mapping is available at https://www.mmv.org/mmv-open/malaria-box/malaria-box-supporting-information. The box was stored at −20 C. Biochemical reagents were purchased from Sigma (St. Louis, MO, USA). The culture media were from ThermoFisher Scientific (formerly Gibco; Waltham, MA, USA).

### 4.2. Cell Culture

Human foreskin fibroblasts (HFF), Vero cells and transgenic beta-galactosidase expressing *Neospora caninum* were cultivated as described [32].

### 4.3. Differential Screening and Inhibition Curves

For an initial screen, 390 compounds (1 µM compound/well) were added to human foreskin fibroblasts (HFFs) previously seeded in 96-well-plates. DMSO (0.25%) was included as a solvent control. HFF monolayers were either left uninfected or were infected with 10^3^
*N. caninum* beta-galactosidase expressing tachyzoites per well. After 3 days, *N. caninum* proliferation was assayed by measuring beta-galactosidase activity, and HFF viability was monitored using the Alamar Blue assay. This screen was repeated once for confirmation, yielding identical results. Forty compounds showing values lower than the cut-off of the beta-galactosidase assay were regarded as effective against *N. caninum*, compounds yielding values lower than the cutoff of the Alamar Blue assay were regarded as cytotoxic. Effective compounds without cytotoxicity were regarded as potential hits and were retained for further experiments. The IC_50_ values (inhibitory concentration of 50% of the solvent control value) of selected compounds were determined in quadruplicate wells in the same assay set-up and were calculated as described [33].

### 4.4. Transmission Electron Microscopy

The transmission electron microscopy of *N. caninum*-infected HFF monolayers was done as previously described [34]. Briefly, HFFs were grown to confluency in 6-well tissue culture devices and were infected with freshly isolated *N. caninum* tachyzoites during 2 h at 37 °C/5% CO_2_. After one wash in PBS, treatments with the compounds (1), (2), and (5) were initiated, all at 500 nM. The controls were treated with the corresponding amount of DMSO. After 6, 12, 24, or 48 h, samples were rinsed in PBS, and were fixed in 100 mM of cacodylate buffer pH7.3 containing 2.5% glutaraldehyde for 2 h at room temperature. The cells were scraped and centrifuged at 1000 g for 10 min at 4 °C and, and the resulting pellet was further fixed in the same solution at 4 °C overnight, followed by post-fixation in 2% OsO_4_ in cacodylate buffer for 4 h at 4 °C. Subsequently, the specimens were washed and pre-stained in Uranyless^TM^ (Electron Microscopy Sciences, Hatfield, PA, USA), followed by dehydration in a graded series of ethanol (30%, 50%, 70%, 90%, and 100%). They were finally embedded in Epon 820 epoxy resin over a period of 2 days with 3 resin changes. The resin was polymerized at 65 °C for 24 h, and 80-nm sections were cut on a Reichert and Jung ultramicrotome. The sections were loaded onto 300-mesh copper grids (Plano GmbH, Marburg, Germany) and were stained with Uranyless^TM^ and lead citrate as described [24].

### 4.5. Statistics

To define the cutoff values in the initial screening, compounds were regarded as effective against *N. caninum* when the beta-galactosidase values were lower than the mean values of the controls (8 replicates) minus three times the standard deviation of these control values. Furthermore, compounds were regarded as cytotoxic when the Alamar Blue values were lower than the mean values of the untreated controls (8 replicates) minus three times the standard deviation of these control values. The IC_50_ values were calculated after the logit-log transformation of the relative growth (RG; control = 1) according to the formula ln (RG/(1 − RG)) = a × ln(drug concentration) + b, and the subsequent regression analysis was carried out using the corresponding software tool contained in the Excel software package (Microsoft, Seattle, WA, USA).

## Figures and Tables

**Figure 1 molecules-25-01460-f001:**
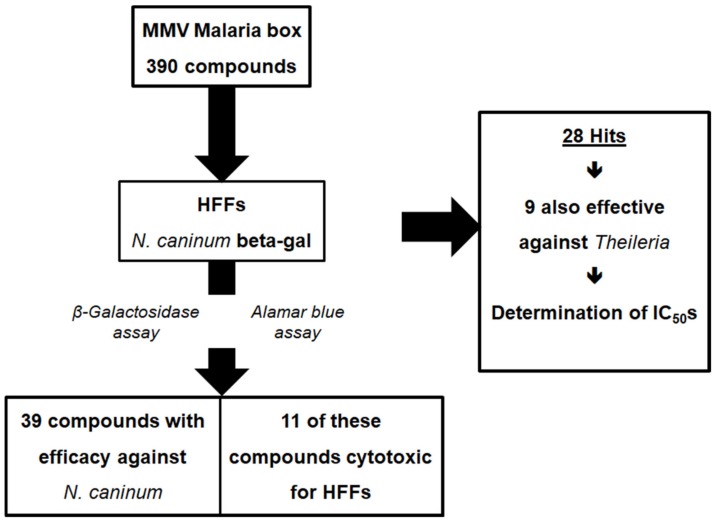
Scheme of the initial screening procedure of 390 Medicines for Malaria Venture (MMV) Malaria Box compounds against *Neospora caninum* tachyzoites and human foreskin fibroblasts (HFFs). The screening was performed as described in Section 4.3. The results of the screening carried out in parallel against *Theileria annulata* were published earlier [18].

**Figure 2 molecules-25-01460-f002:**
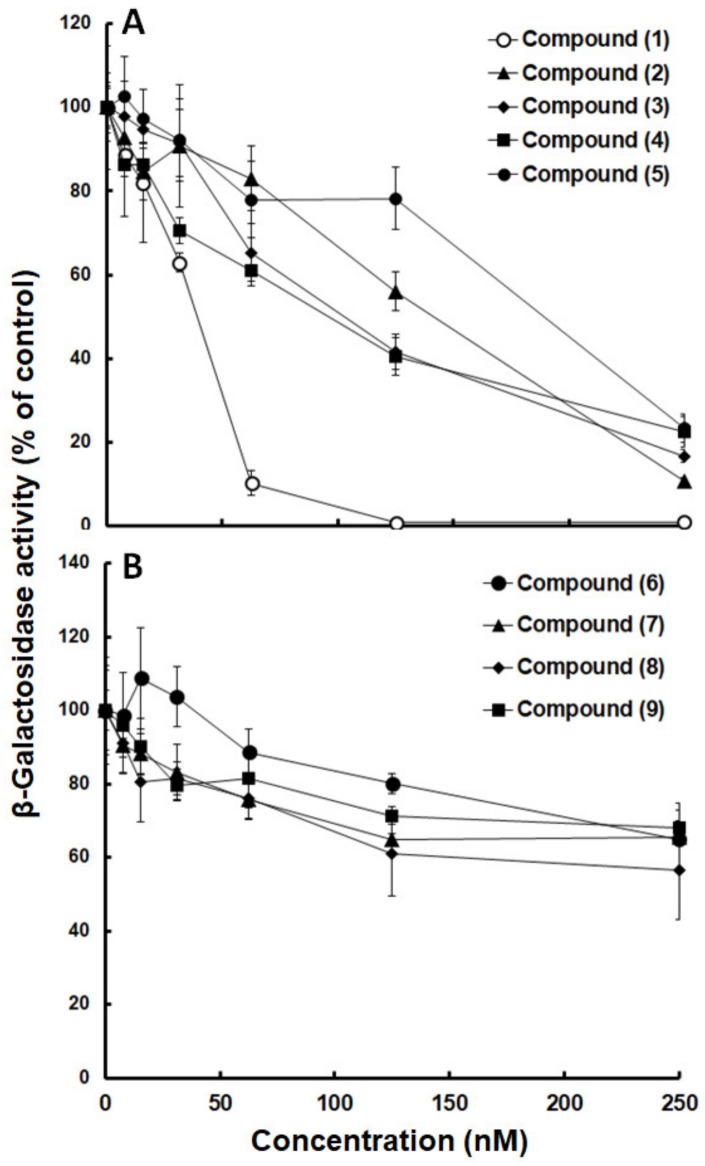
Dose response curves of 9 Medicines for Malaria Venture (MMV) Malaria Box compounds. Compounds 1–5 inhibit *N. caninum* proliferation with IC_50_ values (inhibitory concentration of 50% of the solvent control value) below 200 nM (**A**), compounds 6–9 are less effective (**B**). The inhibition of *N. caninum* tachyzoite proliferation was determined using a β-galactosidase reporter strain in the presence of a concentration series of compounds added prior to the infection of confluent human foreskin fibroblasts by tachyzoites. The mean values ± SE are indicated for quadruplicates.

**Figure 3 molecules-25-01460-f003:**
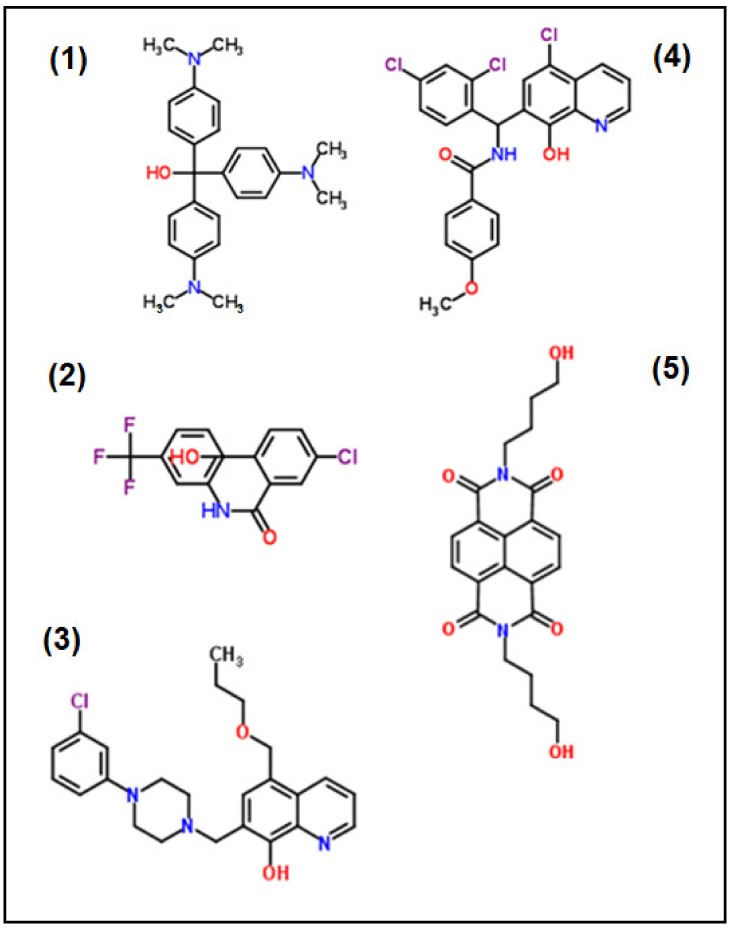
Structures of the five MMV Malaria Box compounds inhibiting *N. caninum* proliferation in vitro with IC_50_ values below 200 nM.

**Figure 4 molecules-25-01460-f004:**
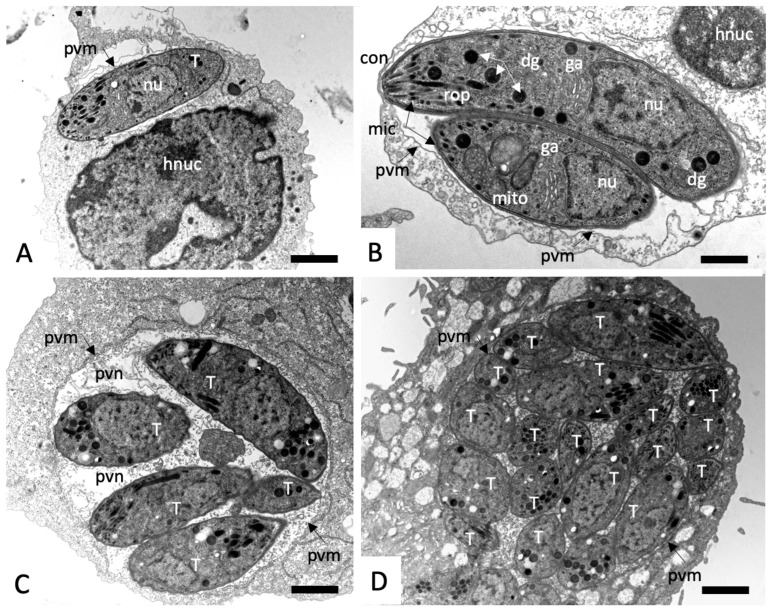
Transmission electron micrographs (TEM) of *N. caninum* tachyzoite-infected HFF fixed and processed at different timepoints (**A**) = 8 h; (**B**) = 14 h; (**C**) = 26 h; (**D**) = 50 h post-infection. con = conoid, dg = dense granules, ga = golgi apparatus, hnuc = host cell nucleus, mic = micronemes, mito = mitochondrion, nu = nucleus, pvm = parasitophorous vacuole membrane, pvn = parasitophorous vacuole network, rop = rhoptries, T = tachyzoites. The bar in (**A**) = 0.8; (**B**) = 0.6 µm; (**C**,**D**) = 0.8 µm.

**Figure 5 molecules-25-01460-f005:**
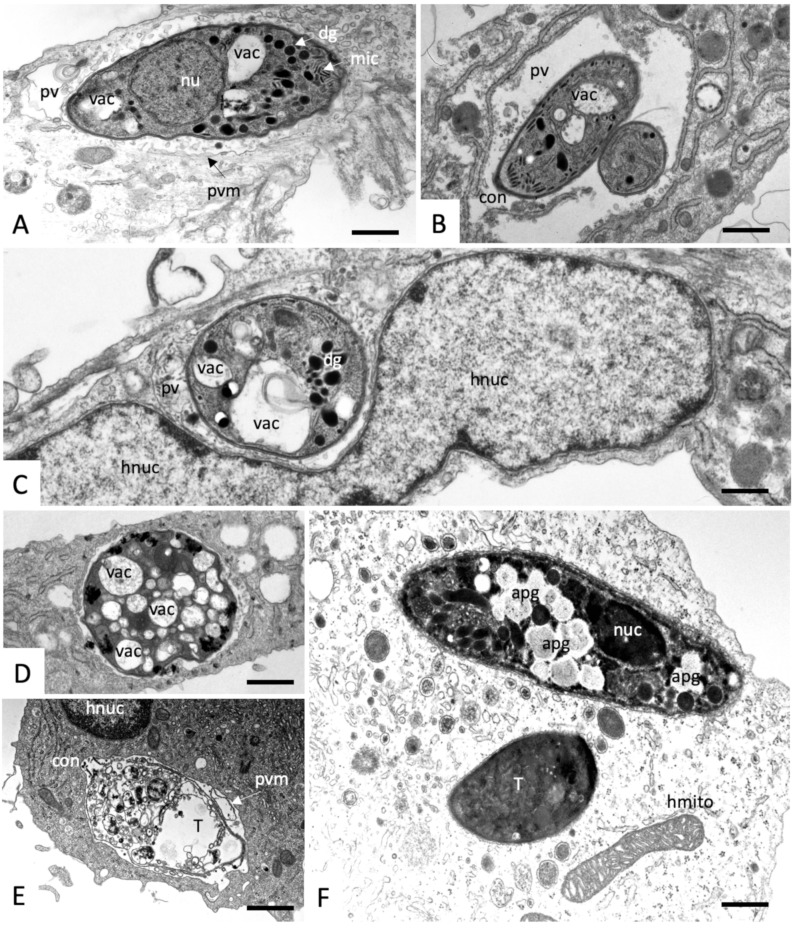
TEM of *N. caninum* tachyzoite-infected HFF treated with a 500 nM compound (1) and fixed and processed at different timepoints after the treatment began (**A**,**B**) = 6 h; (**C**) = 12 h; (**D**) = 24 h; (**E**,**F**) = 48 h. apg = amylopectin granules, dg = dense granules, hmito = host cell mitochondrion, hnuc = host cell nucleus, mic = micronemes, pv = parasitophorous vacuole, pvm = parasitophorous vacuole membrane, T = tachyzoite, vac = vacuolization. The bar in (**A**) = 0.6 µm; (**B**) = 0.8 µm; (**C**) = 0.4 µm; (**D**) = 0.5 µm; (**E**,**F**) = 0.8 µm.

**Figure 6 molecules-25-01460-f006:**
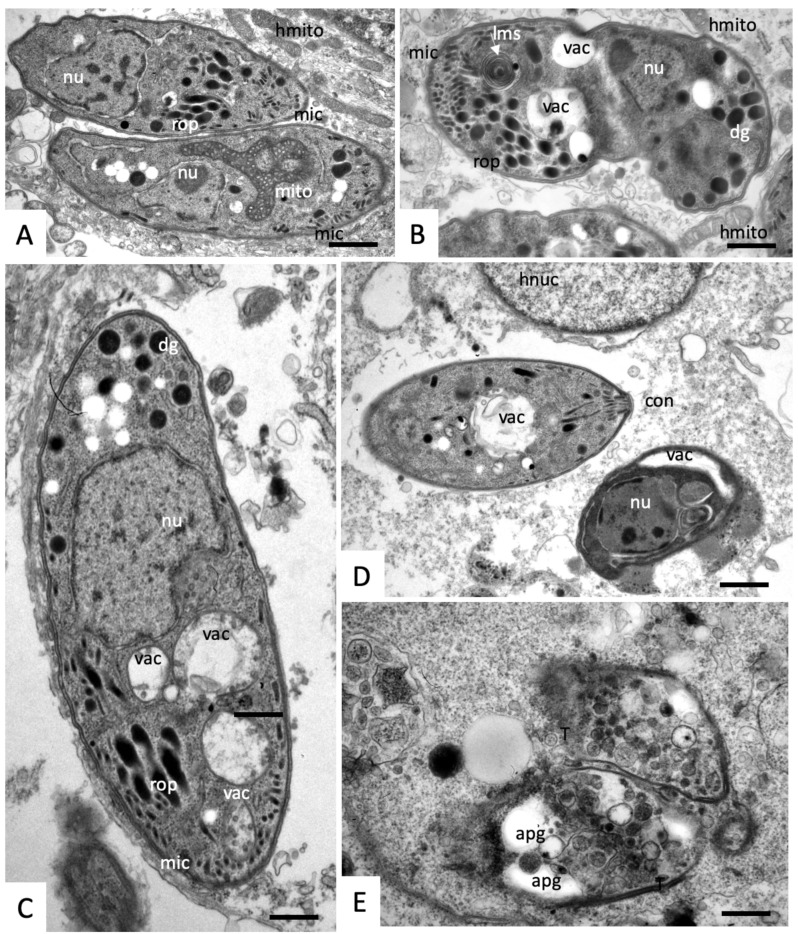
TEM of N. caninum tachyzoite-infected HFF treated with a 500-nM compound (2) and fixed and processed at different timepoints after the treatment began (**A**,**B**) = 6 h, (**C**) = 12 h, (**D**) = 24 h, (**E**) = 48 h). apg = amylopectin granules, con= conoid, dg = dense granules, ga = golgi apparatus, hmito = host cell mitochondrion, hnuc = host cell nucleus, mic = micronemes, mito = mitochondrion, nu = nucleus, pvm = parasitophorous vacuole membrane, pvn = parasitophorous vacuole network, rop = rhoptries, T = tachyzoites, vac = vacuolization. The bar in (**A**,**B**) = 0.6 µm; (**C**) = 0.3 µm; (**D**) = 0.5 µm; (**E**) =0.4 µm.

**Figure 7 molecules-25-01460-f007:**
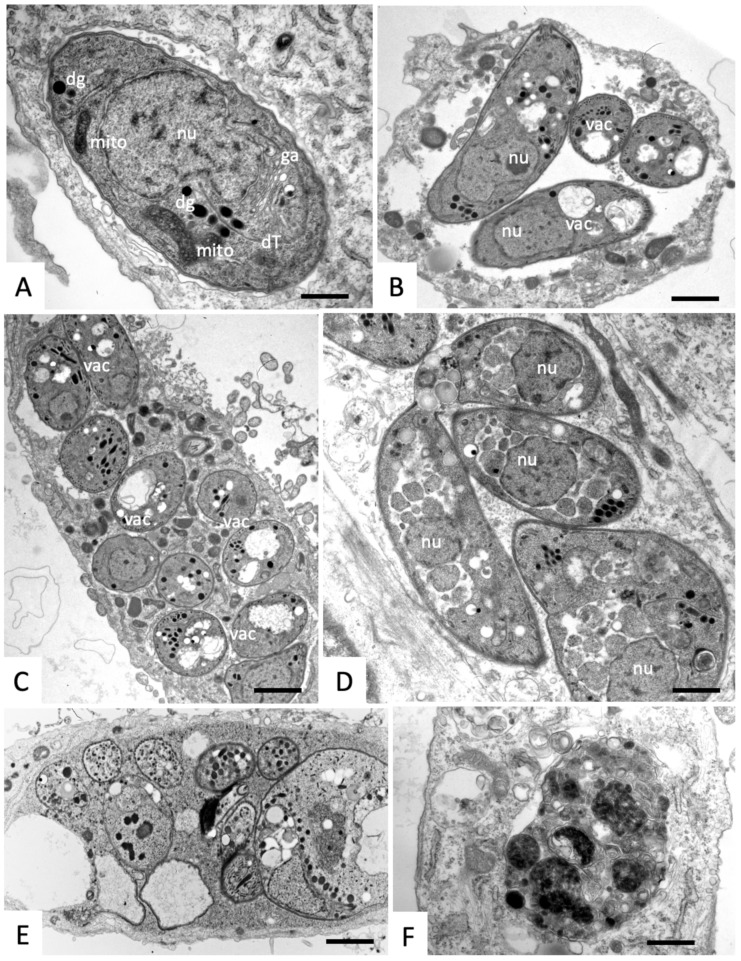
TEM of *N. caninum* tachyzoite-infected HFF treated with a 500-nM compound (5) and fixed and processed at different timepoints after the treatment began (**A**,**B**) = 6 h, (**C**) = 12 h, (**D**) = 24 h, (**E**,**F**) = 48 h). dg = dense granules, dT = daughter zoites, ga = golgi apparatus, mito = mitochondrion, nu = nucleus, vac = vacuolization. The bar in (**A**) = 0.4 µm; (**B**,**C**) = 1 µm; (**D**) = 0.6 µm; (**E**) = 1.2 µm; (**F**) = 0.5 µm.

**Table 1 molecules-25-01460-t001:** Overview of the nine most effective MMV Malaria Box compounds. The IC_50_s were determined from the inhibition curves depicted in Figure 2 as described in Section 4.4. The IC_50_ values were calculated after the logit-log transformation of the mean values and are given in nM with 95% confidence intervals. *, values that were extrapolated. Sources: SJ, St. Jude; G, GlaxoSmithKline; N, Novartis.

Compound	MMV-N°	ChEMBL_NTD_ID	Source	LogP	IC_50_ (nM)
(1)	665941	SJ000140980	SJ	4.76	25 ± 5
(2)	665807	SJ000010289	SJ	6.06	71 ± 30
(3)	000787	SJ000117451	SJ	4.71	99 ± 10
(4)	666054	TCMDC-125549	G	6.15	79 ± 10
(5)	009085	GNF-Pf-3184	N	−1.30	150 ± 30
(6)	000788	SJ000117455	SJ	4.53	280 ± 80 *
(7)	666080	TCMDC-123701	G	4.64	550 ± 120 *
(8)	006522	GNF-Pf-2807	N	6.06	310 ± 110 *
(9)	006172	TCMDC-123912	G	3.50	430 ± 100 *

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
