# Peer review of "In Vitro Activities of MMV Malaria Box Compounds against the Apicomplexan Parasite Neospora caninum, the Causative Agent of Neosporosis in Animals"

_molecules, 2020, doi:10.3390/molecules25061460_

Round 1

Reviewer 1 Report

This interesting study refers about the activity of 3 out of 5 from MMV Malaria Box compounds effective against tachyzoite of Neospora caninum in cell cultures, assessed by electron microscopy. THese compounds are suggested as promising potential candidates for the in vivo treatment of infected calves. The paper is very interesting and worth of publishing in its present form.

Author Response

Thank you for the positive review

Reviewer 2 Report

The manuscript entitled ‘In vitro activities of MMV Malaria Box compounds  against the apicomplexan parasite Neospora caninum, the causative agent of neosporosis in animals’ submitted to Molecules by Hemphill and coworkers have stated the importance of developing compounds that will inhibit the proliferation of N. Caninum tachyzoites which is responsible for the economic losses due to abortion in cattle upto 1.3 billion dollars.

They have identified the compounds showing good IC50 activities and some of those were studied further which led to the analysis that along with the inhibition such compounds also cause the alteration in mitochondria. These are promising compounds for the further studies.

In my opinion the manuscript is very well written, and results have been presented logically and should be accepted for publication.

Few minor errors were spotted in the 4.4 section where 37 C should be replaced by 37 °C, CO2 and OsO4 should be written as CO2 and OsO4, 4 hrs should be replaced by 4 h.

Author Response

Thank you for your positive review. The errors were corrected (corrections in red font).

Reviewer 3 Report

Mueller et al. describe a screen of MMV Malaria Box compounds against Neospora caninum. Nine ‘hit’ compounds were probed further with the IC50 determined for all 9 and drug inhibitory phenotypes over time explored by electron microscopy. The paper is well written and clear. Discussion of the compounds and their activity against other parasites is well done. The experiments and data presented are appropriate to such a study. The TEM data provide some interesting data on phenotypic changes over time with drug treatment for 3 compounds. To identify the drug targets in N. caninum will require substantial additional work and is outside the scope of this well written paper. I have only one concern in terms of fully disclosing the number of replicates across experiments, otherwise, a very nice study.

Major comments:

-Its not very clear where replicates were done or not. In Fig 2, quadruplicates were mentioned. Is this wells or biological replicates. How many times was the initial screen done at 1uM?

Author Response

Thank you for your positive review. We have added information to the section 4.3 to clarify the missing points:

For an initial screen, 390 compounds (1 µM compound / well) were added to human foreskin fibroblasts (HFFs) previously seeded in 96-well-plates. DMSO (0.25%) was included as a solvent control. HFF monolayers were either left uninfected or were infected with 103 N. caninum beta-galactosidase expressing tachyzoites per well. After 3 days, N. caninum proliferation was assayed by measuring beta-galactosidase activity, and HFF viability was monitored using the Alamar Blue assay. This screen was repeated once for confirmation, yielding identical results. Forty compounds showing values lower than the cut-off of the beta-galactosidase assay were regarded as effective against N. caninum, compounds yielding values lower than the cutoff of the Alamar Blue assay were regarded as cytotoxic. Effective compounds without cytotoxicity were regarded as potential hits and were retained for further experiments. IC50 values (inhibitory concentration of 50% of the solvent control value) of selected compounds were determined in quadruplicate wells in the same assay set-up, and were calculated as described [33].
